# CD3ζ-Mediated Signaling Protects Retinal Ganglion Cells in Glutamate Excitotoxicity of the Retina

**DOI:** 10.3390/cells13121006

**Published:** 2024-06-08

**Authors:** Rui Du, Ping Wang, Ning Tian

**Affiliations:** 1Department of Ophthalmology and Visual Science, University of Utah School of Medicine, Salt Lake City, UT 84132, USA; u6034319@utah.edu (R.D.); ping.wang@utah.edu (P.W.); 2Department of Neurobiology, University of Utah, Salt Lake City, UT 84132, USA; 3Department of Biomedical Engineering, University of Utah, Salt Lake City, UT 84132, USA; 4Veterans Affairs Medical Center, Salt Lake City, UT 84148, USA

**Keywords:** RGC protection, CD3ζ, glutamate excitotoxicity, immune proteins, retinal, SFK, Src, Syk

## Abstract

Excessive levels of glutamate activity could potentially damage and kill neurons. Glutamate excitotoxicity is thought to play a critical role in many CNS and retinal diseases. Accordingly, glutamate excitotoxicity has been used as a model to study neuronal diseases. Immune proteins, such as major histocompatibility complex (MHC) class I molecules and their receptors, play important roles in many neuronal diseases, while T-cell receptors (TCR) are the primary receptors of MHCI. We previously showed that a critical component of TCR, CD3ζ, is expressed by mouse retinal ganglion cells (RGCs). The mutation of CD3ζ or MHCI molecules compromises the development of RGC structure and function. In this study, we investigated whether CD3ζ-mediated molecular signaling regulates RGC death in glutamate excitotoxicity. We show that mutation of CD3ζ significantly increased RGC survival in NMDA-induced excitotoxicity. In addition, we found that several downstream molecules of TCR, including Src (proto-oncogene tyrosine-protein kinase) family kinases (SFKs) and spleen tyrosine kinase (Syk), are expressed by RGCs. Selective inhibition of an SFK member, Hck, or Syk members, Syk or Zap70, significantly increased RGC survival in NMDA-induced excitotoxicity. These results provide direct evidence to reveal the underlying molecular mechanisms that control RGC death under disease conditions.

## 1. Introduction

Retinal ganglion cells (RGCs) relay visual signals from the eyes to the brain. The death of RGC in retinal diseases, such as glaucoma and traumatic optic neuropathy (TON), leads to permanent blindness. Numerous studies suggest that multiple mechanistic processes control the death of RGCs under various disease conditions. Glutamate-induced excitotoxicity triggers an increase in the intracellular calcium concentration with several other cellular and molecular mechanisms that have been attributed to both glaucomatous RGC death [1,2,3,4,5,6,7,8,9] and RGC death in TON [7,10,11,12]. However, the roles of many of these pathogenesis processes in RGC death have not been characterized. Further understanding of the molecular mechanisms of RGC death in these diseases will provide crucial information for the development of novel treatment strategies.

Many reports have shown that major histocompatibility complex I (MHCI) molecules and their receptors play critical roles in the pathogenesis of the CNS and retina. Mice with defective MHCI or its receptors have abnormal RGC axonal projections in the brain, abnormal synaptic connections in the visual cortex, abnormal motor neuron function, excessive loss of synaptic connections, and axonal regeneration after injury [13,14,15,16,17,18]. In the retina, MHCI, the key components of the T-cell receptor (TCR) complex, such as the CD3 complex (cluster of differentiation 3), and the major downstream cascades of the TCR complex, such as Src (proto-oncogene tyrosine-protein kinase) family kinases (SFK) and spleen tyrosine kinase (Syk), are expressed by RGCs [14,19,20,21,22,23]. The genetic mutation of MHCI or CD3 ζ-chain (CD3ζ) resultes in significant defects in the development of RGC structure and function [14,19]. In addition, activation of MHCI promotes locomotor abilities after spinal cord injury [13,17,18,24] while inactivation of MHCI lessens brain injury after stroke [25]. Therefore, these immune proteins could regulate the normal structure and function as well as the pathogenesis of neurons through MHCI or its receptors in the brain and retina.

Excessive activation of glutamate receptors leads to neuronal damage and death, and it plays a crucial role in many CNS diseases [26,27] and retinal diseases, including glaucoma, diabetic retinopathy, optic nerve injury, and retinal ischemia [1,3,8,9,28,29]. Accordingly, glutamate excitotoxicity has been widely used as a model to study neuronal death in various diseases. In this study, we characterize the effects of the CD3ζ mutation and the inhibitors of SFK or Syk on RGC death caused by glutamate excitotoxicity. We show that RGC survival is increased significantly in CD3ζ mutants with glutamate excitotoxicity. Also, intraocular injection of inhibitors of SFK or Syk significantly increases RGC survival in glutamate excitotoxicity. Further, pharmacological experiments demonstrate that CD3ζ regulates RGC death through a molecular pathway mediated by an SFK member, Hck, and Syk members, Syk, and Zap70 (ζ-chain-associated protein kinase). These results provide critical evidence to reveal the underlying molecular mechanisms that control RGC death and the molecular targets to develop treatment strategies for RGC protection under disease conditions.

## 2. Materials and Methods

### 2.1. Animals

Wild-type (WT, Strain #:000664) and B6.129S4-Cd247tm1Lov/J (CD3ζ-/-, Strain #:002704) mice are both from a C57BL/6J background and were initially obtained from The Jackson Laboratory (Bar Harbor, ME, USA). They were further bred and maintained at the animal facility of the University of Utah, Moran Eye Center. All animal care was performed following protocols approved by the IACUC of the University of Utah and the IACUC of the VA Salt Lake City Health Care System in compliance with PHS guidelines and those prescribed by the Association for Research in Vision and Ophthalmology (ARVO, Rockville, MD, USA).

### 2.2. Intraocular Injection of N-methyl-d-Aspartic Acid (NMDA) and Inhibitors

The glutamate receptor agonist, NMDA (Sigma, Burlington, MA, USA, Cat#: M3262), was injected into the eyes of both WT and CD3ζ-/- mice to induce glutamate excitotoxicity. The procedure of intraocular injection has been described previously (Figure 1A) [9,19]. In this study, the dosages of NMDA used to cause 50% RGC death in WT mice (3.13 nmol) were determined in a previous study [9,30]. This dosage was injected into the eyes in a 2 μL solution. The distribution of the solution inside the eyes was confirmed by co-injecting NMDA with Alexa Fluor^TM^ 488 conjugated Cholera Toxin Subunit B (CTB, 0.2%, Cat #: C22841, Thermo Fisher Scientific, Eugene, OR, USA), and the retinas were examined by imaging the distribution of the fluorescent signaling of Alexa Fluor^TM^ 488 (Figure 1C).

To determine whether CD3ζ regulates RGC death through SFK and Syk family kinases and which SFK and Syk family members are involved in the singling, four SFK inhibitors (PP2, A419259, SU6656, and Saracatinib) [31,32,33,34,35,36,37,38] and one Syk/Zap70 inhibitor (Piceatannol) [39,40] were used in the study. The names, sources, IC_50_, and references of the inhibitors used in this study are listed in Table 1.

To reduce the impact of the variation of RGC density among mice, we injected 2 μL of NMDA solution into one eye and used the non-injected contralateral eyes as controls to calibrate the RGC survival rate of each mouse. In preparation for intraocular injection, the mice were anesthetized with Isoflurane (1–5% Isoflurane mixed with room air delivered at a rate between 0.8 and 0.9 L/min) through a mouse gas anesthesia head holder (David KOPF Instruments, Tujunga, CA, USA). The 0.5% proparacaine hydrochloride ophthalmic solution was locally applied to each eye. Glass micropipettes made from borosilicate glass using a Brown-Flaming horizontal puller were used for intraocular injection. The glass needles were mounted on a Nano-injection system (Nanoject II, Drummond Scientific Company, Broomall, PA, USA), which precisely controlled the amount of injected solution at the nanoliter (nl) level. The glass needles were aimed at penetrating the eyeball near its equator under a stereo microscope. After the injection, the eyes were covered with 0.5% erythromycin ophthalmic ointment, and the mice were placed in a clean cage sitting on a water blanket. The temperature of the water blanket was set at 33 °C. Mice in this cage were continuously monitored until they completely recovered from anesthesia, and then they were returned to their home cages. The procedures for anesthesia and intraocular injection were approved by the IACUC of the University of Utah and the IACUC of the VA Salt Lake City Health Care System.

### 2.3. Primary Antibodies

Multiple antibodies were used to label RGCs [41,42], SFK members [43,44,45,46,47,48,49], and Syk/Zap70 [50,51,52] family members. The names, sources, methods of validation, concentrations, and references of the antibodies are listed in Table 2.

### 2.4. Preparation of Retinal Whole-Mounts and Retinal Section for Antibody Staining

RGCs were imaged on whole-mount retinal preparation for cell density measurements and slice preparation for specific protein expression. The procedures for immuno-labeling retinal neurons on retinal whole-mount and slide preparations have been described previously [9,19,30]. In brief, mice were euthanized with 100% CO_2_, followed by cervical dislocation.

For whole-mount retinal preparation, retinas were isolated and fixed in 4% paraformaldehyde (PFA) in 0.01M phosphate-buffered saline (PBS; pH 7.4) for 60 min at room temperature. Fixed retinas were washed 10 min × 3 times in 0.01 M PBS and incubated in blocking solution (10% normal donkey serum) at 4 °C for 2 h. Next, retinas were incubated with a guinea pig polyclonal anti-RBPMS antibody (PhosphoSolutions, Aurora, CO, USA) [41,42] (1:500) for 7 days at 4 °C. A Cyanine CyTM 3-conjugated donkey anti-guinea pig (1:400) secondary antibody (Jackson ImmunoResearch, West Grove, PA, USA) was used for 48 h at 4 °C to reveal anti-RBPMS antibody staining (Figure 1B). After the antibody incubation, the retinas were washed 10 min × 3 times and incubated in DAPI (4′,6-diamidino-2-phenylindole, 1 μg/mL, Sigma, Burlington, MA, USA, Cat#: D9542) overnight at 4 °C. Then, the retinas were further washed 10 min × 3 times and flat-mounted on Super-Frost slides (Cat#: 12-550-143, Fisher Scientific, Pittsburgh, PA, USA) with Vectashield mounting medium (Cat#: H-1000, Vector Laboratories, Burlingame, CA, USA).

For retinal section preparation, the whole eyes were removed and fixed in 4% paraformaldehyde (PFA) for 1 h. Fixed eyes were washed 10 min × 3 times in 0.01 M PBS, moved to a 15% sucrose solution for 1.5 h at room temperature, and then incubated in 30% sucrose at 4 °C overnight. Fixed eyes were then embedded in Tissue-Tek OCT compound (Cat#: 4583, Sakura Finetek USA, Torrance, CA, USA) and stored at −80 °C until they were ready for sectioning. Whole eyes were sectioned vertically with a thickness of 12–15 μm. using a Leica CM-3050S cryostat microtome (Leica Biosystems, Wetzlar, Germany), and collected on Super-Frost Plus slides (Cat#: 12-550-15, Fisher Scientific, Pittsburgh, PA, USA). Various antibodies were used to label RGCs, SFK, and Zap70/Syk family members (overnight at 4 °C), and corresponding secondary antibodies (2 h at room temperature) were used to reveal the binding of primary antibodies (Table 2). Then retinal sections were incubated in DAPI (1 μg/mL) for 30 min at room temperature.

### 2.5. Confocal Laser Scanning Microscopy and Image Sampling

The image acquisition and processing have been described in our previous studies [9,19,30,53]. Images of the retina were collected using a dual-channel Zeiss confocal microscope (Axio Examiner D1, Carl Zeiss AG, Jena, Germany) with a C-Apochromat 40′ 1.2 WKorr water immersion lens. Fiji Image J (Version: 2.14.0/1.54f, National Institutes of Health, Bethesda, MD, USA) [54] was used to align multi-stacks of images together, quantify the number of RGCs, and adjust the intensity and contrast of images.

It has been widely reported that the susceptibilities of RGCs to injuries vary significantly among different RGC types. The distribution of RGC types and density also vary with eccentricity and the location of the retina. To avoid the variations in RGC density due to eccentricity and retinal location among different groups, we scanned four stacks of images at four-quarters of each retina, 600 μm away from the center of the optic nerve head of all mice (Figure 1A). Each stack covered 304 μm × 304 μm of the retina and the entire thickness of the ganglion cell layer (GCL) in whole-mount retinas at intervals of 0.5 μm. The density of anti-RBPMS antibody-labeled RGCs in each retina was averaged from the four stacks.

### 2.6. Statistical Analysis

The data are all presented as mean ± SE in the text and figures. Student *t*-tests are used to examine the difference between two means from different animals, and paired *t*-tests are used to compare two means from different eyes (left eyes vs. right eyes) of the same mice.

## 3. Results

### 3.1. CD3ζ Mutation Reduces RGC Death in Glutamate Excitotoxicity

It is well demonstrated that both the TCR-mediated adaptive immune system and complement-mediated innate immune system regulate neuronal development and pathogenesis in the CNS [1,13,14,15,16,17,18,19,25,55,56]. We have previously shown that CD3*ζ* is expressed by RGCs and displaced amacrine cells in the mouse retina [19], and mutation of CD3*ζ* impairs dendritic development of RGCs and ACs [19,53]. Because the MHCI mutation in mice alters neuronal survival in the CNS [13,17,18,25], we thought to determine if CD3*ζ* and its downstream molecules regulate RGC death in the retina. Accordingly, we injected 3.13 nmol NMDA into the left eyes of both WT and age-matched CD3*ζ-/-* mice. In our previous study, intraocular injection of 3.13 nmol NMDA caused approximately 40–50% RGC death within 24 h [30]. Twenty-four hours after the NMDA injection, we collected the retinas from both the left (NMDA-treated) and right (control) eyes, labeled the RGCs using an anti-RBPMS antibody, and imaged and quantified the RGC density of the retinas (Figure 1A). Figure 1B shows a representative image of a whole-mount retina labeled with an anti-RBPMS antibody. A previous study demonstrated that the anti-RBPMS antibody labels all RGCs [54]. Figure 1C shows that the injected solution with CTB-Alexa Fluor^TM^ 488 is evenly distributed on the entire retina. Figure 1D,E show representative images of the retinas of WT and CD3*ζ-/-* mice without NMDA injection. The RGC density of the WT retina (4947 ± 115 cells/mm^2^ (average ± SE)) is not different from that of CD3*ζ-/-* mice (5039 ± 140 cells/mm^2^, Student *t*-test, *p* = 0.679, Figure 1H) without NMDA injection. Figure 1F,G show representative images of the retinas of WT and CD3*ζ-/-* mice 24 h after 3.13 nmol NMDA injection. The RGC density of the WT mice was reduced from 4947 ± 115 cells/mm^2^ to 2540 ± 136 cells/mm^2^ (paired *t*-test, *p* = 0.0003, n = 5) and the RGC density of the CD3*ζ-/-* mice was reduced from 5039 ± 140 cells/mm^2^ to 3380± 201 cells/mm^2^ (paired *t*-test, *p* < 0.0001, n = 5) 24 *h* after NMDA injection (Figure 1H). However, the RGC density of CD3*ζ-/-* mice with NMDA injection is significantly higher than that of WT mice with NMDA injection (2540 ± 136 cells/mm^2^ for WT vs. 3380± 201 cells/mm^2^ for CD3*ζ-/-* mice, Student *t*-test, *p* = 0.017, Figure 1H), which is 1.33 folds of WT mice. These results demonstrated that the CD3*ζ* mutation partially but significantly increases RGC survival in NMDA excitotoxicity.

### 3.2. Hck Is Expressed by RGCs in Mouse Retina

In T-cells, the MHCI complex is commonly recognized by the TCR complex. The engagement of MHCI with TCR activates several downstream molecular cascades through the CD3*ζ*-SFK-Zap70/Syk signal pathway (Figure 2A) [57]. In vertebrates, eight “typical” SFKs (Src, Blk, Fgr, Fyn, Hck, Lck, Lyn, and Yes) and four “atypical” SFKs (Yrk, Brk, Frk, and Srm) have been identified [58,59,60,61]. Among these SFK members, seven of them (Src, Fyn, Lyn, Lck, Yes, Fgr, and Yrk) are expressed in the retina [20,21,22,23,62,63,64].

In this study, we confirmed the expression of these SFK members in the mouse retina. Figure 2B–E show representative images of a retinal cross-section of a WT mouse co-labeled with an anti-RBPMS antibody, various anti-SFK antibodies (Src, Fyn, Lck, or Yes, respectively), and DAPI. Further, we found an additional SFK member, Hck, is expressed in the mouse retina (Figure 2F). Therefore, the mouse retina expresses seven SFKs (Src, Fyn, Lyn, Lck, Yes, Fgr, and Hck) [20,21,22,23,62,63,64] except Yrk, which is only expressed in the chicken retina [20] (please see negative controls of the antibody staining in Appendix A). In addition to RGCs, several SFKs, such as Src, Fyn, and Lck, are also expressed by other retinal neurons (Figure 2B–D). Furthermore, these SFKs are also expressed in the retinas of CD3*ζ-/-* mice (Appendix A).

### 3.3. Selective Inhibition of Hck Increases RGC Survival in NMDA-Induced Excitotoxicity

In CNS neurons, SFKs act as important signaling intermediaries, regulating a variety of outputs, such as cell proliferation, differentiation, apoptosis, migration, and metabolism. In the retina, it is unclear which SFK regulates RGC survival/death through CD3*ζ*. To identify the SFKs responsible for the CD3*ζ*-mediated RGC death, we pharmacologically dissected the contribution of the seven SFKs in mouse RGC death due to NMDA-induced excitotoxicity using four SFK inhibitors (PP2, A419259, SU6656, and Saracatinib). Because each of these four SFK inhibitors selectively inhibits 4–7 SFKs, together they can inhibit eight SFKs, including all seven SFKs expressed in the mouse retina (Table 1).

Figure 3A shows the experimental procedure of pharmacological dissection of the contribution of the SFKs in mouse RGC death due to NMDA-induced excitotoxicity. In these experiments, one of the four SFK inhibitors was injected into the right eye of each WT mouse on day 1. The concentration of the inhibitors is determined as 100× of the IC_50_s, which are obtained from published studies (Table 1). One hour after the injection of an inhibitor, NMDA was injected into both the left (NMDA) and right (NMDA and inhibitor) eyes of the mice. Twenty-four hours later, the retinas of both eyes were collected and fixed, labeled with an anti-RBPMS antibody, imaged using a confocal microscope, and the RGC density was quantified. Figure 3B–E show representative images of the retina of WT mice without intraocular injection (Figure 3B), with intraocular injection of NMDA (Figure 3C), with intraocular injection of PP2 and NMDA (Figure 3D), and with intraocular injection of A419259 and NMDA (Figure 3E). Quantitatively, the RGC densities of eyes treated with PP2 and NMDA are 1.7 fold of their opposite eyes, only with NMDA injection of the same five mice (2259 ± 96 cells/mm^2^ for NMDA vs. 3765 ± 119 cells/mm^2^ for PP2 and NMDA eyes, paired *t*-test, *p* < 0.0001, Figure 3F), while the RGC densities of eyes treated with A419259 and NMDA are 1.4 fold of their opposite eyes, only with NMDA injection of the same five mice (2294 ± 80 cells/mm^2^ for NMDA vs. 3143 ± 139 cells/mm^2^ for A419259 and NMDA eyes, paired *t*-test, *p* = 0.002, Figure 3G). These results demonstrated that both PP2 and A419259 increase RGC survival in NMDA-induced excitotoxicity. Because PP2 and A419259 inhibit four and five different SFKs, respectively, we proposed that the SFKs inhibited by both PP2 and A419259 are the ones that protect RGCs from NMDA-induced excitotoxicity. As shown in Figure 3H, PP2 inhibits Src, Lck, Hck, and Fyn, while A419259 inhibits Src, Lck, Hck, Lyn, and Fgr, and they both inhibit Src, Lck, and Hck. Therefore, we conclude that PP2 and A419259 likely protect RGCs by inhibiting Src, Lck, or Hck.

To determine whether Src, Lck, and Hck all participate in NMDA-induced RGC death, we tested two additional SFK inhibitors, SU6656 and Saracatinib, which inhibit five or six SFKs, including Src and Lck but not Hck (Table 1). Figure 4A–D show representative images of the retina of WT mice without intraocular injection of NMDA (Figure 4A), with intraocular injection of NMDA (Figure 4B), intraocular injection of SU6656 and NMDA (Figure 4C), and intraocular injection of Saracatinib and NMDA (Figure 4D). Quantitatively, the RGC densities of retinas treated with SU6656 or Saracatinib before NMDA injection are not different from those of eyes treated with NMDA alone (Figure 4E,F). These results demonstrated that none of the seven SFKs inhibited by SU6656 and Saracatinib, including Src and Lck, protect RGCs from NMDA-induced excitotoxicity (Figure 4G). Combining these results with those from Figure 3, we conclude that the SFKs increasing RGC survival in NMDA-induced excitotoxicity have to be sensitive to both PP2 and A419259 but insensitive to both SU6656 and Saracatinib and Hck is the only SFK that meets this criterion (Figure 5). Therefore, PP2 and A419259 protect RGCs in NMDA-induced excitotoxicity by inhibiting Hck.

It has been reported that some SFK inhibitors could regulate other signal pathways in addition to SFK-mediated signal pathways. For instance, PP2 and A419259 could not only regulate RGC survival in NMDA excitotoxicity by inhibiting Hck-mediated signal pathways in our study, but they have also been reported to inhibit pathways mediated by EGFR/PI3K–AKT or PKC/TrkB pathways for neuronal survival [31,32,33,34,65]. To further determine whether PP2 and A419259 intermediate RGC death through CD3*ζ* independent signal pathways, we examined the effect of PP2 and A419259 on RGC survival in NMDA-induced excitotoxicity of CD3*ζ-/-* mice. If PP2 and A419259 increase RGC survival by inhibiting the CD3*ζ*-mediated pathway, they would not have an additional effect on RGC survival in CD3*ζ-/-* mice. Figure 6A–D show representative images of the retina of CD3*ζ-/-* mice without intraocular injection (Figure 6A), with intraocular injection of NMDA (Figure 6B), intraocular injection of PP2 and NMDA (Figure 6C), and intraocular injection of A419259 and NMDA (Figure 6D). Surprisingly, the RGC densities of retinas treated with PP2 or A419259 before NMDA injection are significantly lower than those of eyes treated with NMDA alone. Quantitatively, the RGC densities of eyes treated with NMDA and PP2 and NMDA are 3329 ± 151 cells/mm^2^ and 2589 ± 85 cells/mm^2^, respectively (n = 5, paired *t*-test, *p* = 0.001, Figure 6E). The RGC densities of eyes treated with NMDA and A419259 and NMDA are 3126 ± 111 cells/mm^2^ and 2325 ± 111 cells/mm^2^, respectively (n = 5, paired *t*-test, *p* = 0.0002, Figure 6F). These results demonstrated that not only did PP2 and A419259 not increase RGC survival in NMDA-induced excitotoxicity of CD3*ζ-/-* mice, but they also promoted additional RGC death in these mice. Therefore, both PP2 and A419259 are likely to inhibit other signal pathways that normally protect RGCs in addition to SFK-mediated pathways. To maximize the effect of inhibition of the CD3*ζ*-mediated mechanism on RGC protection, it is necessary to identify additional molecular targets that will allow the inhibition of CD3*ζ*-mediated RGC death without intermediating other pathways that counteract the protective effects.

### 3.4. Syk/Zap70 Family Kinases Are Expressed by RGCs in Mouse Retina and Mediate RGC Death

In T-cells, the activation of CD3*ζ*-mediated cascades requires the involvement of both SFKs and Syk/Zap70 family kinases (Figure 7A) [57]. While SFKs regulate a variety of signaling pathways, Syk/Zap70 family kinases are much more specifically associated with the CD3*ζ*-chain. Therefore, inhibition of Syk/Zap70 family kinases will likely provide a more specific effect through a CD3*ζ*-Syk/Zap70-mediated mechanism on RGC death. To determine whether Syk/Zap70 family kinases are required for CD3*ζ*-mediated RGC death, we first examined whether Syk/Zap70 family kinases were expressed by RGCs in WT mice. Accordingly, we co-labeled the retinal cross-sections of WT mice with anti-Syk antibody (green), anti-RBPMS antibody (red), and DAPI (blue) (Figure 7B). A zoom-in view of the area in the dish-line box of Figure 7B shows that the anti-Syk staining in GCL completely overlaps with the anti-RBPMS staining. Figure 7C shows a representative image of a retinal cross-section of a WT mouse co-labeled with an anti-Zap70 antibody (green), an anti-RBPMS antibody (red), and DAPI (blue). Similarly, a zoom-in view of the area in the dish-line box of Figure 7C shows that the anti-Zap70 staining in GCL (Figure 7C1) and the anti-RBPMS staining (Figure 7C2) completely overlap with each other (Figure 7C4). Also, we labeled the Syk and Zap70 in CD3*ζ-/-* mouse retina. Figure 7D,E show that both Syk and Zap70 are expressed by RGCs in CD3*ζ-/-* mouse retina.

We then tested whether the inhibition of Syk/Zap70 family kinases increases RGC survival in NMDA-induced excitotoxicity. Accordingly, we intraocularly injected a Syk/Zap70 family kinase inhibitor, piceatannol [39,40], following the experimental procedure described in Figure 3A. Figure 8A–C shows representative images of the retina of WT mice without intraocular injection (Figure 8A), with intraocular injection of NMDA (Figure 8B), and with intraocular injection of piceatannol and NMDA injection (Figure 8C). Quantitatively, the RGC density of retinas treated with piceatannol injection before NMDA (3248 ± 135 cells/mm^2^) is significantly higher than that of eyes treated with NMDA alone (2141 ± 64 cells/mm^2^, paired *t*-test, n = 5 for both groups, *p* = 0.0004, Figure 8D), demonstrating that piceatannol increases RGC survival in NMDA-induced excitotoxicity.

To further determine whether piceatannol intermediates RGC only through the CD3*ζ*-Syk/Zap70 pathway, we examined the effect of piceatannol on RGC survival in NMDA-induced excitotoxicity on CD3*ζ-/-* mice, in which the CD3*ζ*-Syk/Zap70 pathway is already inactivated by CD3*ζ* mutation. Figure 8E–G shows representative images of the retina of CD3*ζ-/-* mice without intraocular injection (Figure 8E), with intraocular injection of NMDA (Figure 8F), and with intraocular injection of piceatannol before NMDA injection (Figure 8G). Quantitatively, the RGC density of retinas treated with piceatannol before NMDA injection (3331 ± 90 cells/mm^2^) is not different from that of eyes treated with NMDA alone (3423 ± 51 cells/mm^2^, paired *t*-test, n = 5 for both groups, *p* = 0.267, Figure 8H). These results demonstrated that piceatannol has no additional effect on RGC survival in CD3*ζ-/-* mice, indicating it only intermediates with the CD3*ζ*-Syk/Zap70 pathway in RGC protection.

## 4. Discussion

In this study, we show that the mutation of CD3ζ significantly enhanced RGC survival in NMDA-induced excitotoxicity, supporting the idea that CD3ζ-mediated signaling regulates RGC death in retinal diseases. Second, our study revealed that Hck is expressed by RGCs, and SFK inhibitors sensitive to Hck protect RGCs in NMDA-induced excitotoxicity. Third, the downstream molecules of CD3ζ-SFKs, Syk/Zap70, are also expressed by RGCs, and the Syk/Zap70 inhibitor protects RGCs from NMDA-induced excitotoxicity. The protective effects of Hck inhibitors and Syk/Zap70 inhibitors are abolished in CD3ζ mutants. These results demonstrated that CD3ζ regulates RGC death through the CD3ζ-Hck-Syk/Zap70 signaling pathway. Finally, although genetic deletion of CD3ζ or pharmacological inhibition of the Hck-Syk/Zap70 signaling pathway significantly increased the survival of RGCs in NMDA-induced excitotoxicity, neither the genetic nor the pharmacological approach completely rescues all RGCs. These results strongly support the idea that the CD3ζ-Hck-Syk/Zap70 signaling pathway is only responsible for the survival of a sub-group of RGCs in NMDA-induced excitotoxicity.

### 4.1. How Does Glutamate Excitotoxicity Damage RGCs in Retinal Diseases?

Glutamate excitotoxicity is thought to play a critical role in RGC death in many retinal diseases, such as glaucoma, diabetic retinopathy, optic nerve injury, retinal ischemia, and several other retinal diseases [2,3,4,5,6,56]. One proposed mechanism for glaucomatous RGC death attributes the elevated IOP to glutamatergic excitotoxicity. Consistently, elevated IOP increases the expression of the NMDA receptor (NMDAR) in DBA/2J mice [8], and the number of NMDAR-positive RGCs is reduced parallel to the loss of RGC in a model with chronic, elevated IOP [28]. In addition, an NMDA antagonist, memantine, significantly reduces RGC loss and the expression of NMDARs [66,67], suggesting that NMDARs are involved in RGC death in glaucoma. Furthermore, elevated IOP activates NMDARs, which trigger mitochondria-mediated apoptosis through the release of optic atrophy 1 (OPA1) [68]. Blockade of glutamate receptors inhibits OPA1 release, increases Bcl-2 expression, decreases Bax expression, and blocks apoptosis in the glaucomatous mouse retina [69]. In diabetic retinopathy (DR), RGC injury occurs before microvascular damage via multiple mechanisms, including overstimulation of the NMDAR [70,71,72]. Consistently, there is an elevated glutamate level in aqueous humor and vitreous in DR animal models and DR patients [73,74]. In addition, the immunoreactivities of NR1 and GluR2/3 are upregulated in RGCs of both patients with diabetes and experimental DR animals [75,76], and blocking of NMDAR protects RGCs against neurodegeneration in DR rats [77].

Excessive stimulation of glutamate receptors, such as the NMDARs, can cause excitotoxicity by allowing high levels of calcium ions (Ca^2+^) to enter the neurons [78]. Ca^2+^ influx into cells activates many enzymes, including phospholipases, endonucleases, and proteases. These enzymes can damage cell structures such as the cytoskeleton, cell membrane, and DNA [79,80]. Also, high-level calcium influx through NMDARs can shut off cAMP response element binding (CREB) protein, which in turn causes apoptosis and cell death [81], and NMDARs are expressed by all RGCs [82,83].

In addition to NMDARs, calcium-permeable AMPA receptors (CP-AMPARs) might also mediate glutamate excitotoxicity [84,85]. CP-AMPARs mediate fast excitatory synaptic transmission and a rapid influx of extracellular Ca^2+^ [86,87]. In the retina, CP-AMPAR is expressed by all RGCs [88,89,90,91]. Following NMDA exposure or elevated IOP, RGCs rapidly upregulate CP-AMPAR expression [88,92,93,94]. Blocking CP-AMPARs reduces Ca^2+^ levels in RGCs [89,92,93,94], and increases RGC survival [92,93]. Together, these findings indicate a key role of glutamate receptors in the increase in intracellular Ca^2+^ levels after ocular injury, consistent with findings elsewhere in the nervous system [95].

### 4.2. How do SFKs Regulate NMDA-Induced RGC Death through the CD3ζ-Hck-Syk/Zap70 Signal Pathway?

Our results demonstrated that the genetic mutation of CD3ζ significantly enhanced RGC survival in NMDA-induced excitotoxicity, and pharmacological inhibition of Hck or Syk/Zap70 protects RGCs in NMDA-induced excitotoxicity in a CD3ζ-dependent manner. These results support the idea that CD3ζ regulates RGC death through the CD3ζ-Hck-Syk/Zap70 signaling pathway. A fundamental question is, what are the underlying molecular mechanisms that enable CD3ζ-Hck-Syk/Zap70 signaling to regulate NMDA-induced excitotoxicity and RGC death?

NMDARs are multi-subunit complexes consisting of scaffolding proteins, kinases, phosphatases, and trafficking machinery that regulate ionotropic modulation, intracellular signaling, and synaptic stabilization [96]. Phosphorylation and dephosphorylation of NMDAR subunits are major mechanisms of regulating NMDAR function [97,98,99], and the SFKs are key regulators of NMDARs [99,100,101]. Consistently, Fyn is found to be a component of the NMDAR complex in many areas of the CNS [102,103] and can phosphorylate NMDAR [104]. Activation of Fyn increases NMDAR activity [105,106] and SFK-mediated NMDAR activity is increasingly implicated in normal physiology and diverse disease pathologies. For instance, the potentiation of NMDAR activity by SFKs has been described in the hippocampus, prefrontal cortex, and spinal cord [107,108,109]. Intracellular stimulation of Src potentiates NMDAR activity and occludes NMDAR-dependent LTP [110], while blockade of endogenous Src activity suppresses NMDAR activity and induction of LTP in hippocampal neurons [109,111]. In addition, upregulation of NMDAR activity escalates the sensitivity of chronic pain [58], and this hyperfunction of NMDAR is thought to be regulated by both Src and Fyn [112,113,114]. However, our pharmacological experiments demonstrated that neither Src nor Fyn seem to be involved in the RGC death due to NMDA-mediated excitotoxicity, although both Src and Fyn are expressed by RGCs. Therefore, it needs to be further examined whether Hck regulates NMDAR activity in NMDA-induced excitotoxicity through similar mechanisms as Src and Fyn.

One might ask which cell/ligand may be responsible for activating CD3ζ on RGCs during retinal degeneration. In immune cells, such as T-cells, CD3ζ is a critical component of the T-cell receptor (TCR) complex, and MHCI functions as the primary ligand of TCR. In the retina, MHCI is expressed by RGCs and displaced amacrine cells. Genetic mutation of MHCI results in phenotypic defects on the RGC axons closely resembling those of CD3ζ mutation [14,19]. Therefore, MHCI might be responsible for activating CD3ζ on RGCs during retinal degeneration. In addition to regulating NMDARs, the activation of CD3ζ could trigger several downstream molecular cascades, including mobilizing intracellular calcium and reorganizing the cytoskeleton [57]. These pathways could also participate in RGC degeneration. However, the ligand-receptor relationship between MHCI and TCR in the retina has not been well established.

The roles of MHCI and its receptors in the pathogenesis of the CNS remain contradictory. Several reports showed that MHCI-deficient mice have abnormal retinogeniculate connections, abnormal motor learning, abnormal synaptic plasticity in the visual cortex, reduced regeneration of axons, and extensive loss of synapses on motor neurons after injury [13,14,15,16,17,18,19], while up-regulated MHCI expression significantly promoted the recovery of locomotor abilities after spinal cord injury in mice [24]. In another report, mice with MHCI and PirB knockouts had smaller infarcts and enhanced motor recovery in a stroke model, less cell death after ischemia of the hippocampus, and a reduced reactive astrocytic response after middle cerebral artery occlusion [25]. Thus, the activation of MHCI and its receptors appears to promote the recovery of motor neurons after spinal cord injury but exacerbates brain injury in ischemia after stroke [25]. In our studies, the genetic mutation of CD3ζ increases RGC survival in NMDA-induced excitotoxicity, which is similar to the effects in CNS neurons [25]. However, mutation of CD3ζ does not prevent RGC death after optic nerve crush (ONC) [53], suggesting that RGC death induced by NMDA excitotoxicity and ONC are likely regulated by different molecular signaling pathways. Consistently, several mechanisms have been proposed to address how MHCI and its receptors regulate neuron survival after injuries. For instance, MHCI and its receptors likely regulate motor neuron survival after spinal cord transection through a signal pathway within motor neurons [13,24,115], while the death of motor neurons in amyotrophic lateral sclerosis is thought to be the result of astrocyte-induced toxicity [24]. Also, the effects of MHCI and its receptors could be cell-type-specific. This is supported by the reports that activation of MHCI and its receptors on motor neurons stabilize synaptic connections, increase synapse formation, and limit secondary neuronal degeneration after spinal cord transection [13,24,115], while activation of MHCI and PirB of CNS neurons limits their axonal outgrowth in regeneration after brain ischemia induced by stroke [25,116,117,118,119]. In addition, different types of pathological insults might trigger different effects mediated by MHCI and its receptors. In PirB mutants, ischemia triggers an increase in the number of midline crossing fibers from the undamaged corticospinal tract into the denervated red nucleus [25], while spinal cord injury or traumatic brain injury does not affect axonal regeneration, functional recovery, or axonal plasticity of neurons [120,121]. Although both PirB and TCR function as the receptors of MHCI [122], PirB is not expressed in the retina. Therefore, TCR, or CD3ζ, will most likely be the receptor for MHCI in the retina. However, whether CD3ζ is activated by the engagement of MHCI with TCR in the retina needs to be further investigated.

### 4.3. What Are the Regulatory Mechanisms of CD3ζ-Mediated RGC Death in Retinal Diseases?

Although pharmacologically inhibiting the SFKs/Syks or genetically mutating the CD3ζ significantly reduced RGC death in NMDA-induced excitotoxicity, neither of these interferences completely rescued RGCs. Therefore, our results suggest that CD3ζ-Hck-Syk/Zap70 signaling might only mediate the death of some RGC types in NMDA-induced excitotoxicity. Consistently, the type-specific susceptibility of RGCs has been reported in many studies. For instance, it has been shown that the susceptibility of RGCs to glutamate excitotoxicity depends on soma size and retinal eccentricity. Larger RGCs at the peripheral retina are more sensitive to kainite-induced excitotoxicity; smaller RGCs at the central retina are more sensitive to NMDA-induced excitotoxicity [123], while melanopsin-expressing RGCs (ipRGCs) are most resistant to NMDA-induced excitotoxicity [124,125]. In animal models of glaucoma, RGCs with a large somata, big axon, or OFF response appear to be more vulnerable to elevated IOP [126,127,128,129,130,131,132], while ON RGCs are more susceptible to elevated IOP than ON-OFF RGCs [133]. In addition, the transient OFF αRGCs exhibited a higher rate of cell death, while neither sustained OFF αRGCs nor sustained ON αRGCs have reduced synaptic function due to elevated IOP [132]. Similar to models with elevated IOP, OFF RGCs are more susceptible than ON RGCs to ONC, and ON sustained RGCs seem to be more susceptible than ON transient RGCs [134]. Among αRGCs, ipRGCs, DSRGCs, and W3-RGCs, αRGCs seem to be the least susceptible type to ONC [135]. We recently compared the susceptibility of αRGCs, W3-RGCs, BD-RGCs, and J-RGCs to NMDA-induced excitotoxicity and ONC. Among these four RGC types, αRGCs are the least susceptible RGC type to NMDA-induced excitotoxicity, while both BD-RGCs and J-RGCs are the most sensitive RGC types to NMDA-induced excitotoxicity [9]. On the other hand, BD-RGCs and αRGCs are the least susceptible RGC types to ONC, and W3-RGCs are the most sensitive RGC types [30]. Furthermore, our previous studies showed that mutation of CD3ζ has no effect on RGC death with ONC [53]. These results strongly support the idea that multiple molecular mechanisms participate in the regulation of RGC death in an RGC type-specific and injury-specific manner.

## 5. Conclusions

In conclusion, we investigated the roles of CD3ζ and the inhibitors of SFKs or Syks on RGC death in glutamate excitotoxicity. We show that mutations of CD3ζ significantly increased RGC survival in NMDA-induced excitotoxicity. In addition to the previously reported six SFKs and two Syk kinases (Syk/Zap70), we found Hck is expressed in mouse RGCs. Inhibition of Hck or Syks increased RGC survival in NMDA-induced excitotoxicity. Further, pharmacological studies demonstrated that CD3ζ regulates RGC death through a molecular signaling pathway mediated by the CD3ζ-Hck-Syk/Zap70 pathway. However, neither pharmacological inhibition of Hck/Syks nor genetic mutation of CD3ζ completely rescues RGCs from NMDA-induced excitotoxicity. In combination with our previous studies, these results support the idea that multiple molecular mechanisms participate in the regulation of RGC death in an RGC type-specific and injury-specific manner. These results provide critical insights for understanding the underlying molecular mechanisms that control RGC death and the development of a strategy to prevent RGC death under disease conditions.

## Figures and Tables

**Figure 1 cells-13-01006-f001:**
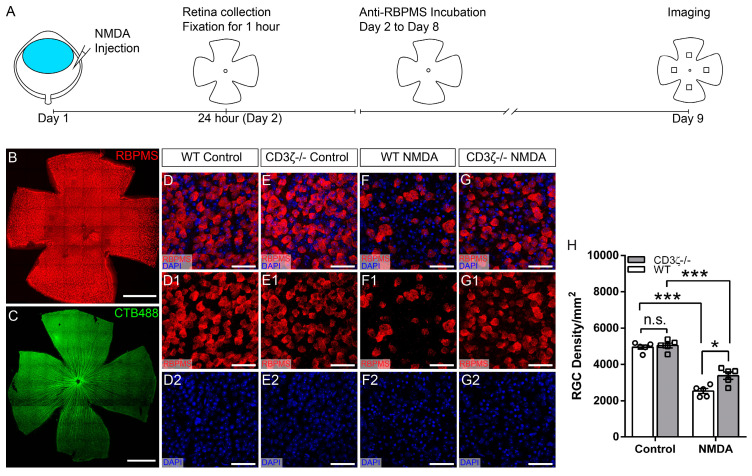
CD3*ζ* mutation reduces RGC death due to glutamate excitotoxicity. (**A**) A schematic view of the experimental procedure. (**B**) A representative image of a flat-mount retina labeled with an anti-RBPMS antibody. (**C**) A representative image of a flat-mount retina labeled by Alexa Fluor^TM^ 488 conjugated Cholera Toxin Subunit B. (**D**) Representative images of flat-mount retina of a wild-type (WT) mouse without NMDA injection show the overlapping of the staining of anti-RNA-binding protein with multiple splicing (RBPMS) antibody (red) and DAPI (4′,6-diamidino-2-phenylindole) (blue) (**D**), anti-RBPMS staining (**D1**), and DAPI staining (**D2**). (**E**) Representative images of the flat-mount retina of a CD3*ζ* mutant mouse (CD3*ζ-/-*) without NMDA injection show the overlapping of the staining of anti-RBPMS and DAPI (E), anti-RBPMS staining (**E1**), and DAPI staining (**E2**). (**F**) Representative images of the flat-mount retina of a WT mouse 24 h after NMDA injection show the overlapping of the staining of anti-RBPMS and DAPI (**F**), anti-RBPMS staining (**F1**), and DAPI staining (**F2**). (**G**) Representative images of the flat-mount retina of a CD3*ζ-/-* mouse 24 *h* after NMDA injection show the overlapping of the staining of anti-RBPMS and DAPI (**G**), anti-RBPMS staining (**G1**), and DAPI staining (**G2**). (**H**) Densities of RGCs labeled with anti-RBPMS antibody in WT and CD3*ζ-/-* mice with (NMDA) or without (control) intraocular NMDA injection (Student *t*-test). n.s., not significant; * *p* < 0.05; *** *p* < 0.001. Each dot indicates an individual eye. n = 5 for each group. Scale bars in panels (**B**,**C**): 1 mm. Scale bars in panels (**D**–**G**): 40 μm.

**Figure 2 cells-13-01006-f002:**
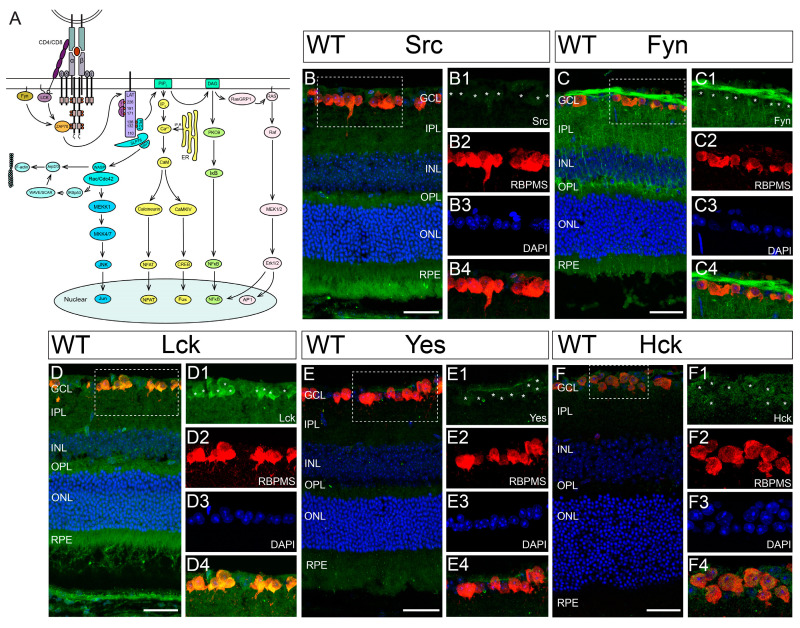
Multiple SFKs are expressed by RGCs in mouse retina. (**A**) A schematic view of molecular arrangement of TCR/CD3*ζ* complex and its downstream molecular cascades in T-cells. (**B**) A representative image of a retinal cross-section of a WT mouse co-labeled with anti-Src antibody (green), anti-RBPMS antibody (red, for RGCs), and DAPI (blue for all nuclei in the retina section). A zoom-in view of the area in the dish-line box of panel B shows the anti-Src staining in RGCs (**B1**), anti-RBPMS staining (**B2**), DAPI staining (**B3**), and the overlapping of the staining of anti-Src, anti-RBPMS, and DAPI (**B4** (**C**) A retinal cross-section of a WT mouse co-labeled with anti-Fyn antibody, anti-RBPMS antibody, and DAPI. (**C1**–**C4**) shows the staining of anti-Fyn, anti-RBPMS, and DAPI, and the overlapping of the staining in the area in the dish-line box of panel (**C**). (**D**) A retinal cross-section of a WT mouse co-labeled with anti-Lck antibody, anti-RBPMS antibody, and DAPI. (**D1**–**D4**) shows the staining of anti-Lck, anti-RBPMS, and DAPI, and the overlapping of the staining in the area in the dish-line box of panel (**D**). (**E**) A retinal cross-section of a WT mouse co-labeled with anti-Yes antibody, anti-RBPMS antibody, and DAPI. (**E1**–**E4**) shows the staining of anti-Lck, anti-RBPMS, and DAPI, and the overlapping of the staining in the area in the dish-line box of panel (**E**). (**F**) A retinal cross-section of a WT mouse co-labeled with anti-Hck antibody, anti-RBPMS antibody, and DAPI. (**F1**–**F4**) shows the staining of anti-Hck, anti-RBPMS, and DAPI, and the overlapping of the staining in the area in the dish-line box of panel (**F**). Scale bars in panels (**B**–**F**): 40 μm. The asterisks in **B1**, **C1**, **D1**, **E1** and **F1** indicate the location of RGC soma.

**Figure 3 cells-13-01006-f003:**
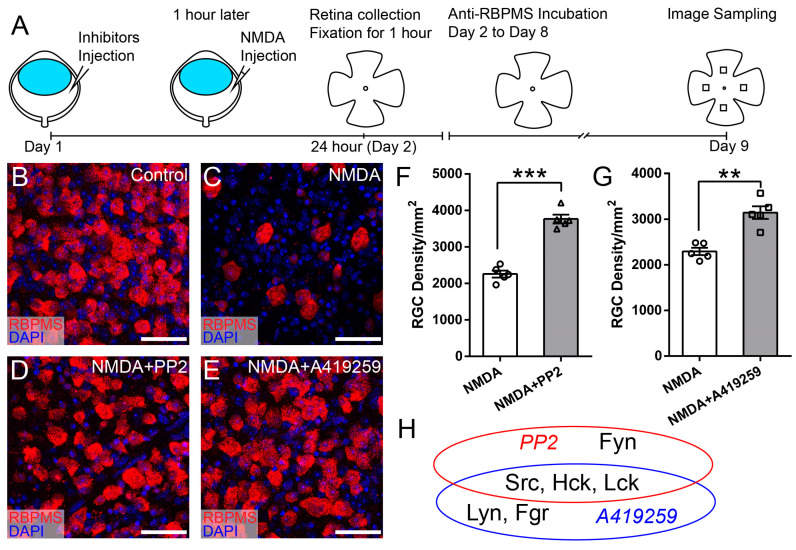
Inhibitors of SFKs, PP2, and A419259, increase RGC survival in NMDA-induced excitotoxicity. (**A**) A schematic view of the experimental procedure of pharmacological dissection of the contribution of the SFK members in mouse RGC death due to NMDA-induced excitotoxicity. (**B**) A representative image of a flat-mount retina of a normal WT mouse with by anti-RBPMS antibodies (red) and DAPI (blue). (**C**) A representative image of a flat-mount retina of a WT mouse 24 h after NMDA intraocular injection. The retina is labeled with an anti-RBPMS antibody (red) and DAPI (blue). (**D**) A representative image of a flat-mount retina of a WT mouse treated by intraocular injection of PP2 and NMDA. The retina was collected 24 h after NMDA injection and labeled with an anti-RBPMS antibody (red) and DAPI (blue). (**E**) A representative image of a flat-mount retina of a WT mouse treated by intraocular injection of A419259 and NMDA. The retina was collected 24 h after NMDA injection and labeled with an anti-RBPMS antibody (red) and DAPI (blue). (**F**) Densities of RGCs labeled with an anti-RBPMS antibody in the left eyes with NMDA intraocular injection (NMDA) and the right eyes with intraocular injection of PP2 one hour before NMDA injection (NMDA and PP2) of the same mice (n = 5, paired *t*-test). (**G**) Densities of RGCs labeled with an anti-RBPMS antibody in the left eyes with NMDA intraocular injection (NMDA) and the right eyes with intraocular injection of A419259 1 h before NMDA injection (NMDA and A419259) of the same mice (n = 5, paired *t*-test). In panels (**F**) and G, ** 0.001 < *p* < 0.01, *** *p* < 0.001. Each dot indicates an individual eye. (**H**) A schematic view of the SFKs sensitive to PP2, A419259, and both. Scale bars in panels (**B**–**E**): 40 μm.

**Figure 4 cells-13-01006-f004:**
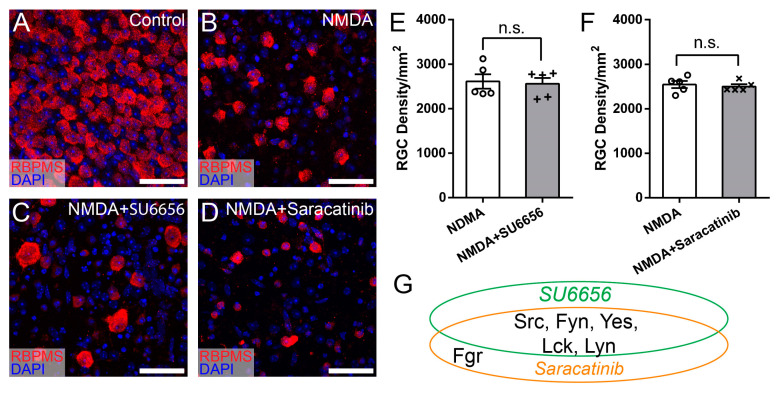
Inhibitors of SFKs, SU6656 and Saracatinib, did not increase RGC survival in NMDA-induced excitotoxicity. (**A**) A representative image of a flat-mount retina of a normal WT mouse labeled with an anti-RBPMS antibody (red) and DAPI (blue). (**B**) A representative image of a flat-mount retina of a WT mouse 24 h after NMDA intraocular injection. The retina is labeled with an anti-RBPMS antibody (red) and DAPI (blue). (**C**) A representative image of a flat-mount retina of a WT mouse treated by intraocular injection of SU6656 and NMDA. The retina was collected 24 h after NMDA injection and labeled with an anti-RBPMS antibody (red) and DAPI (blue). (**D**) A representative image of a flat-mount retina of a WT mouse treated by intraocular injection of Saracatinib and NMDA. The retina was collected 24 h after NMDA injection and labeled with an anti-RBPMS antibody (red) and DAPI (blue). (**E**) Densities of RGCs labeled with an anti-RBPMS antibody in the retina of left eyes with NMDA intraocular injection (NMDA) and the right eyes with intraocular injection of SU6656 1 h before NMDA injection (NMDA and SU6656) of the same mice (2611 ± 162 cells/mm^2^ for NMDA vs. 2560 ± 131 cells/mm^2^ for SU6656 and NMDA eyes, n = 5, paired *t*-test, *p* = 0.68). (**F**) Densities of RGCs labeled with an anti-RBPMS antibody in the left eyes with NMDA intraocular injection (NMDA) and the right eyes with intraocular injection of Saracatinib 1 h before NMDA injection (NMDA and Saracatinib) of the same mice (2543 ± 80 cells/mm^2^ for NMDA vs. 2500 ± 49 cells/mm^2^ for Saracatinib and NMDA eyes, n = 5, paired *t*-test, *p* = 0.683). In panels (**E**,**F**), each dot indicates an individual eye. n.s. no significance. (**G**) A schematic view of the SFKs sensitive to SU6656 and Saracatinib. Scale bars in panels (**A**–**D**): 40 μm.

**Figure 5 cells-13-01006-f005:**
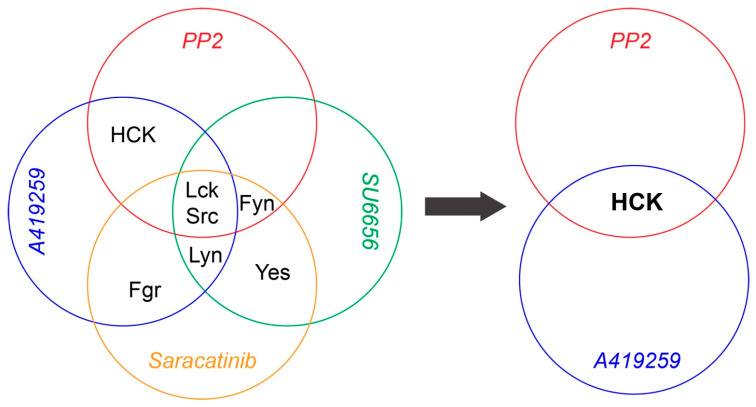
Pharmacological dissection of the SFKs regulating RGC survival in NMDA excitotoxicity. A schematic view of the pharmacological dissection of the SFKs regulating RGC survival in NMDA excitotoxicity. Hck is the only SFK that is sensitive to both PP2 and A419259 but not sensitive to both SU6656 and Saracatinib.

**Figure 6 cells-13-01006-f006:**
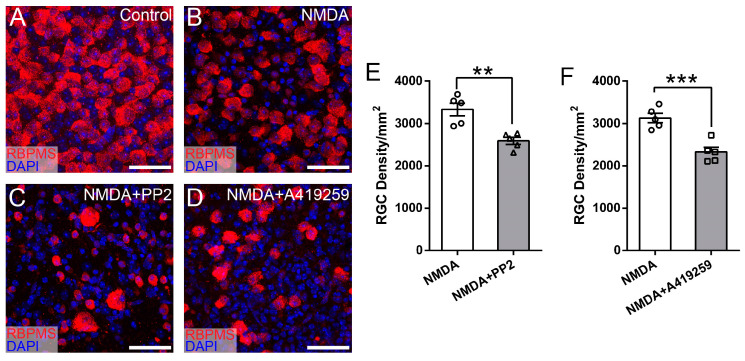
PP2 and A419259 promote RGC death by NMDA excitotoxicity in CD3*ζ-/-* mice. (**A**) A representative image of a flat-mount retina of a CD3*ζ-/-* mouse without intraocular injection of NMDA (control) labeled with an anti-RBPMS antibody (red) and DAPI (blue). (**B**) A representative image of a flat-mount retina of a CD3*ζ-/-* mouse 24 h after NMDA intraocular injection. (**C**) A representative image of a flat-mount retina of a CD3*ζ-/-* mouse treated by intraocular injection of PP2 and NMDA. The retina was collected 24 h after NMDA injection and labeled with an anti-RBPMS antibody (red) and DAPI (blue). (**D**) A representative image of a flat-mount retina of a CD3*ζ-/-* mouse treated by intraocular injection of A419259 and NMDA. The retina was collected 24 h after NMDA injection and labeled with an anti-RBPMS antibody (red) and DAPI (blue). (**E**) Densities of RGCs labeled with an anti-RBPMS antibody in CD3*ζ-/-* mice with NMDA intraocular injection (NMDA) and with intraocular injection of PP2 1 h before NMDA injection (NMDA and PP2). (**F**) Densities of RGCs labeled with an anti-RBPMS antibody in CD3*ζ-/-* mice with NMDA intraocular injection (NMDA) and with intraocular injection of A419259 1 h before NMDA injection (NMDA and A419259). Scale bars in panels (**A**–**D**): 40 μm. In panels (**E**,**F**), each dot indicates an individual eye. ** 0.001 < *p* < 0.01, *** *p* < 0.001.

**Figure 7 cells-13-01006-f007:**
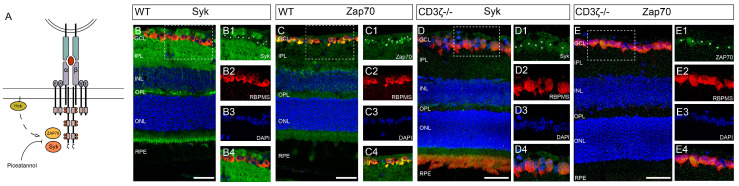
Both Syk and Zap70 are expressed by RGCs in WT and CD3*ζ-/-* mouse retinas. (**A**) A schematic view of how Syk/Zap70 interacts with TCR/CD3*ζ* complex in T-cells and the working site of Piceatannol. (**B**) A representative image of a retinal cross-section of a WT mouse co-labeled with anti-Syk antibody (green), anti-RBPMS antibody (red), and DAPI (blue). A zoom-in view of the area in the dish-line box of panel B shows the anti-Syk staining in RGCs (**B1**), anti-RBPMS staining (**B2**), DAPI staining (**B3**), and the overlapping of the staining of anti-Syk, anti-RBPMS, and DAPI (**B4**). The asterisks in (**B1**) indicate the location of RGC somas labeled in (**B2**). (**C**) A representative image of a retinal cross-section of a WT mouse co-labeled with anti-Zap70 antibody (green), anti-RBPMS antibody (red), and DAPI (blue). A zoom-in view of the area in the dish-line box of panel (**C**) shows the anti-Zap70 staining in RGCs (**C1**), anti-RBPMS staining (**C2**), DAPI staining (**C3**), and the overlapping of the staining of anti-Zap70, anti-RBPMS, and DAPI (**C4**). The asterisks in (**C1**) indicate the location of RGC somas labeled in (**C2**). (**D**) A representative image of a retinal cross-section of a CD3*ζ-/-* mouse co-labeled with anti-Syk antibody (green), anti-RBPMS antibody (red), and DAPI (blue). A zoom-in view of the area in the dish-line box of panel (**D**) shows the anti-Syk staining in RGCs (**D1**), anti-RBPMS staining (**D2**), DAPI staining (**D3**), and the overlapping of the staining of anti-Syk, anti-RBPMS, and DAPI (**D4**). The asterisks in (**D1**) indicate the location of RGC somas labeled in (**D2**). (**E**) A representative image of a retinal cross-section of a CD3*ζ-/-* mouse co-labeled with anti-Zap70 antibody (green), anti-RBPMS antibody (red), and DAPI (blue). A zoom-in view of the area in the dish-line box of panel E shows the anti-Zap70 staining in RGCs (**E1**), anti-RBPMS staining (**E2**), DAPI staining (**E3**), and the overlapping of the staining of anti-Zap70, anti-RBPMS, and DAPI (**E4**). The asterisks in (**E1**) indicate the location of RGC somas labeled in (**E2**). Scale bars in panels (**B**–**E**): 40 μm.

**Figure 8 cells-13-01006-f008:**
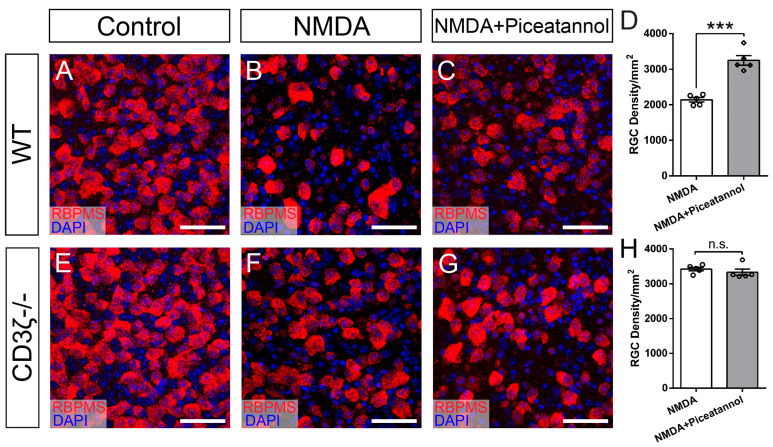
Piceatannol increases RGC survival in NMDA excitotoxicity of WT mice but not CD3*ζ-/-* mice. (**A**–**C**) Representative images of flat-mount retinas of WT mice without intraocular injection (control, (**A**)), 24 h after intraocular injection of NMDA (NMDA, (**B**)), and with intraocular injection of Piceatannol 1 h before NMDA injection (NMDA and Piceatannol, (**C**)). Retinas were co-labeled with an anti-RBPMS antibody (red) and DAPI (blue). (**D**) Densities of RGCs labeled with an anti-RBPMS antibody in WT mice 24 h after intraocular injection of NMDA (NMDA) and with intraocular injection of Piceatannol 1 h before NMDA injection (NMDA and Piceatannol). (**E**–**G**) Representative images of flat-mount retinas of CD3*ζ-/-* mice without intraocular injection (control, (**E**)), 24 h after intraocular injection of NMDA (NMDA, (**F**)), and with intraocular injection of Piceatannol 1 h before NMDA injection (NMDA and Piceatannol, (**G**)). Retinas were co-labeled with an anti-RBPMS antibody (red) and DAPI (blue). (**H**) Densities of RGCs labeled with an anti-RBPMS antibody in CD3*ζ-/-* mice 24 h after intraocular injection of NMDA (NMDA) and with intraocular injection of Piceatannol 1 h before NMDA injection (NMDA and Piceatannol). Scale bars in panels (**A**–**C**) and (**E**–**G**): 40 μm. In panels (**D**,**H**), each dot indicates an individual eye. *** *p* < 0.001, n.s., not significant.

**Table 1 cells-13-01006-t001:** SFK and Syk/Zap70 inhibitors.

Inhibitors	Targeted SFKs	Vendor	Cat #	IC_50_	Dosage	References
PP2	Src, Fyn, Lck, Hck	Santa Cruz Biotechnology, Dallas, TX, USA	sc-202769	4–100 nM	400 nM	[31,35]
A419259	Src, Lck, Lyn, Hck, Fgr,	Sigma, Burlington, MA, USA	SML0446	3–61.8 nM	3.3 mM	[32,33,34]
SU6656	Src, Fyn, Yes, Lyn, Lck	Santa Cruz Biotechnology, Dallas, TX, USA	sc-203286	20 nM–6.88 μM	688 μM	[36]
Saracatinib	Src, Fyn, Yes, Fgr, Lck, Lyn, Blk	Santa Cruz Biotechnology, Dallas, TX, USA	sc-364607	2.7–11 nM	1 μM	[37,38]
Piceatannol	Zap70/Syk	Santa Cruz Biotechnology, Dallas, TX, USA	sc-200610A	10 μM	1 mM	[39,40]

**Table 2 cells-13-01006-t002:** Antibodies used in this study.

Antibody	Antigen	Host	Vendor	Cat#	Validation	Conc.	References
Primary Abs							
Anti-RBPMS	RNA binding protein with multiple splicing	Guinea pig	PhosphoSolutions, Aurora, CO, USA	1832-RBPMS	WB, IHC	1:500	[41,42]
Fyn	Fyn Proto-Oncogene	Goat	Santa Cruz Biotechnology, Dallas, TX, USA	sc-16	WB, IHC	1:100	[43,44]
Lck	Lck proto-oncogene	Rabbit	Santa Cruz Biotechnology, Dallas, TX, USA	sc-28882	WB	1:100	[45]
Hck	Hematopoietic Cell Kinase Hck (phospho Y410)	Rabbit	Abcam, Cambridge, UK	ab61055	WB, IHC	1:200	[46,47]
Src	Src Proto-oncogene	Mouse	Abcam, Cambridge, UK	ab231081	IHC, WB	1:100	[48]
Yes	YES Proto-Oncogene	Rabbit	Cell signaling, Boston, MA, USA	3201S	WB, IP	1:100	[49]
Syk	Spleen tyrosine kinase	Rabbit	Santa Cruz Biotechnology, Dallas, TX, USA	sc-1077	WB, IHC	1:100	[50,51]
Zap70	ζ-chain of T cell receptor-associated protein kinase 70	Rabbit	Santa Cruz Biotechnology, Dallas, TX, USA	sc-574	WB	1:500	[52]
Secondary Abs							
Cyanine CyTM 3-conjugated AffiniPure Donkey Anti-Guinea Pig IgG (H and L)	Guinea pig	Donkey	Jackson ImmunoResearch, West Grove, PA, USA	706-165-148		1:400	
Alexa Fluor^®^ 488 conjugated AffiniPure Donkey Anti-Goat IgG (H and L)	Goat	Donkey	Jackson ImmunoResearch, West Grove, PA, USA	705-545-147		1:400	
Alexa Fluor^®^ 488 conjugated AffiniPure Donkey Anti-Rabbit IgG (H and L)	Rabbit	Donkey	Jackson ImmunoResearch, West Grove, PA, USA	711-545-152		1:400	
Fluorescein (FITC) AffiniPure™ Donkey Anti-Mouse IgG (H+L)	Mouse	Donkey	Jackson ImmunoResearch, West Grove, PA, USA	715-095-151		1:400	

## Data Availability

The data presented in this study are available on request from the corresponding author.

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
