# Peer review of "CD3ζ-Mediated Signaling Protects Retinal Ganglion Cells in Glutamate Excitotoxicity of the Retina"

_cells, 2024, doi:10.3390/cells13121006_

Round 1

Reviewer 1 Report

Comments and Suggestions for Authors

Du et al. reported a study of ‘CD3e-mediated signaling protects RGCs in glutamate excitotoxicity of retina’. They showed that RGCs from CD3e deficient mice were partially rescued from NMDA-induced cell death. Their data suggests that Hck (one of Srcs) and Syk/Zap70 play a key role in NMDA-mediated RGC death. The finding is novel and interesting.

 Major comments:

 1. The data showed knockout of CD3e partially protected NMDA-mediated RGC loss, suggesting CD3e activity plays a role in RGC degeneration. Please discuss which cell/ligand may be responsible for activating CD3e on RGC during retinal degeneration.

 2.     RGC subtypes or in different regions of the retina could have various susceptibilities to injury, as it is mentioned in line 536. In this study, only one stack per quadrant was imaged and used for RGC quantification, which seems less rigorous, especially because RGC densities vary significantly from center to peripheral in the retina. It is important to clarify how exactly the region was determined and selected for RGC quantification or RGC survival from different regions of the retina (central, middle, and peripheral) need to be compared in different treatment or WT/KO mouse groups. 

 3.     In Figure 2, the negative controls for Src, Yes and Hck immunolabeling should be provided as a supplementary figure.

 4.     Please indicate if the Srcs and Syk/Zap70 were also expressed by retinal cells other than RGCs. 

5.     In Figure 7, Syk (green) and RBPMS (red) did not appear to co-localize. The asterisks in B1 were supposed to mark the Syk labeling, but some of them marked an empty space. Please clarify.

Minor comments:

1. Move the information of the number of animals from the bar charts to the figure legends.

 2.     Place the scale bar in the representative image and use a thicker bar. Describe the corresponding length of the scale bar in the figure legend.

 3.     Please add the statistical method to the figure legends.

 4.     The asterisks in bar charts are too small. Please increase the size.

 5.     In the legend of Figure 4, it described * P<0.05, but there is no ‘*’ in Figure 4E. Please clarify.

Author Response

Response to R1

Major comments:

Comment 1. The data showed knockout of CD3z partially protected NMDA-mediated RGC loss, suggesting CD3zactivity plays a role in RGC degeneration. Please discuss which cell/ligand may be responsible for activating CD3z on RGC during retinal degeneration.

Response: This is an important question. Accordingly, we add the following paragraph to the discussion (section 4.2) in this revision.

One might ask which cell/ligand may be responsible for activating CD3z on RGCs during retinal degeneration. In immune cells, such as T-cells, CD3z is a critical component of the T-cell receptor (TCR) complex and MHCI functions as the primary ligand of TCR. In the retina, MHCI is expressed by RGCs and displaced amacrine cells. Genetic mutation of MHCI results in the phenotypic defects on the RGC axons closely resembling that of CD3z mutation [14, 19]. Therefore, MHCI might be responsible for activating CD3z on RGCs during retinal degeneration. In addition to regulating NMDARs, the activation of CD3z could trigger several downstream molecular cascades, including mobilizing intracellular calcium and reorganizing the cytoskeleton [34]. These pathways could also participate RGC degeneration. However, the ligand-receptor relationship between MHCI and TCR in the retina has not been well established.

Comment  2. RGC subtypes or in different regions of the retina could have various susceptibilities to injury, as mentioned in line 536. In this study, only one stack per quadrant was imaged and used for RGC quantification, which seems less rigorous, especially because RGC densities vary significantly from center to peripheral in the retina. It is important to clarify how exactly the region was determined and selected for RGC quantification or RGC survival from different regions of the retina (central, middle, and peripheral) need to be compared in different treatment or WT/KO mouse groups.

Response: It has been widely reported that the susceptibilities of RGCs to injuries vary significantly among different RGC types. The distribution of RGC types and density also vary with eccentricity and location of the retina (such as dorsal vs ventral and temporal vs nasal). Further, it is not clear whether the susceptibilities of RGCs to NMDA excitotoxicity vary at different retinal locations as that in glaucoma patients. Therefore, sampling the RGC density at any selective location of the retina under disease conditions might not precisely represent the responses of all RGC types of the whole retina.

To avoid the variations of RGC density sampling due to eccentricity and location-dependent changes from different mouse groups (WT and transgenic mice with or without injury), we sampled the RGC densities at 600 mm away from the center of the optic nerve head (304 mm x 304 mm, see Figure 1A for details) of four quarters (dorsal, ventral, temporal, and nasal) of all mice, quantified the RGC density of each quarter at this distance, and average the RGC density from all four quarters. As described in the “Material and Method” (section 2.5), image acquisition and processing have been used in our previous studies [9,19,30,31]. Using this approach, our results represent the average RGC densities at 600 mm away from the center of the optic nerve head of four-quarters of all mice to minimize the potential effect of eccentricity and location-dependent changes of RGC density on our results.

Accordingly, we add the following paragraph in the Method (Section 2.5) in this revision to clarify how exactly the region was determined and selected for RGC quantification.

It has been widely reported that the susceptibilities of RGCs to injuries vary significantly among different RGC types. The distribution of RGC types and density also vary with eccentricity and location of the retina. To avoid the variations of RGC density due to eccentricity and retina location among different groups, we scan four stacks of images at four-quarters of each retina at 600 mm away from the center of the optic nerve head of all mice (Figure 1A). Each stack covers 304 mm x 304 mm of the retina and the entire thickness of the ganglion cell layer (GCL) in wholemount retinas at intervals of 0.5 mm. The density of anti-RBPMS antibody labeled RGCs of each retina is averaged from the four stacks.

Comment  3. In Figure 2, the negative controls for Src, Yes and Hck immunolabeling should be provided as a supplementary figure.

Response: According to this comment, the negative controls for the secondary antibodies of donkey anti-rabbit 488, donkey anti-mouse 488, donkey anti-goat 488, and donkey anti-guinea pig Cy3 are provided in a new supplement figure S1 in this revision. These secondary antibodies are used to label Src, Fyn, Yes, Lck, Hck, Syk, and Zap70 in both WT and CD3ζ-/- mouse retinas.

Comment  4. Please indicate if the Srcs and Syk/Zap70 were also expressed by retinal cells other than RGCs. 

Response: Srcs and Syks are also expressed by some other retinal cells as shown in Figs 2B, 2C, and 2D. Accordingly, we add the following sentence to section 3.2 in this revision.

In addition to RGCs, several SFKs, such as Src, Fyn, and Lck, are also expressed by other retinal neurons (Figs 2B-2D).

Comment  5. In Figure 7, Syk (green) and RBPMS (red) did not appear to co-localize. The asterisks in B1 were supposed to mark the Syk labeling, but some of them marked an empty space. Please clarify.

Response: This seems to be a misinterpretation. The anti-Syk antibody preferentially labels the membrane of RGCs but not the cytosol and nuclear of the cells (Fig 7B1 in the revision) and forms a ring-like structure with a hollow center. On the other hand, the anti-RPBMS antibody labels the cytosol of RGCs (Fig 7B2). When Fig 7B1 overlays with Fig 7B2, the ring-like structure in Fig 7B1 overlaps well with the edge of the anti-RPBMS antibody staining of RGCs (Fig 7B and 7B4). However, Fig 7D1 shows more labeling of Syk on the cell membrane and is much better overlapped with the anti-RPBMS antibody staining shown in Fig 7D2.

Minor comments:

Comment  1. Move the information of the number of animals from the bar charts to the figure legends.

Response: Revised accordingly.

Comment  2. Place the scale bar in the representative image and use a thicker bar. Describe the corresponding length of the scale bar in the figure legend.

Response: Revised accordingly.

Comment  3. Please add the statistical method to the figure legends.

Response: Revised accordingly.

Comment  4. The asterisks in bar charts are too small. Please increase the size.

Response: Revised accordingly.

Comment  5. In the legend of Figure 4, it described * P<0.05, but there is no ‘*’ in Figure 4E. Please clarify.

Response: Revised accordingly.

Reviewer 2 Report

Comments and Suggestions for Authors

Summary of the research 

In this study, the authors characterize the effects of CD3ζ mutation and the inhibitors of SFK or Syk on RGC death caused by glutamate excitotoxicity. They primarily used pharmacology and immunofluorescence assays to address these questions.  

The following concerns should be clarified or addressed before the manuscript is accepted. I detail my thoughts and suggestions below

1.     There is a remarkable diversity of RGCs, and the various subtypes have unique morphological features, distinct functions, and characteristic pathways that link the inner retina to the relevant brain areas. The central and peripheral regions of the retina have different numbers of ganglion cells. Provide details on how the consistency of RGC imaging between animals was maintained.

2.     Provide the details of the number of animals, sections/ animals, and region imaged within each section.

3.     RBPMS should label all the retinal ganglion cells? Thus, what is the cell type with only DAPI labeling?  Provide quantification of cells with only DAPI and discussion.

4.     Line 94: The authors used 3.13 nmol NMDA, which does not match the concentration cited in the study (Reference 9), where the concentration is 6.25 nmol. Clarification is needed on how the concentration was determined for this study.

5.     Authors should try different concentrations to determine the optimized NMDA concentrations.

6.     Fig. 1A: According to the text, the experiment was conducted for 24 hours, but the figure indicates an experiment of more than 8 days. Clarification is needed.

7.     Fig. 1B: Authors should include separate panels for DAPI and anti-RBPMS, merging them in a separate panel for better understanding.

8.     Fig. 1C: RGC density is noticeably lower for CD3ζ-/- mice with NMDA, indicating that the CD3ζ mutation cannot protect RGCs from the effects of NMDA, which does not support the objective of this study. Although there appears to be a significant difference between WT and CD3ζ-/- mice, the values are very close (between 2000-3000). I suggest including more mice for a better outcome. Authors should also determine the statistical significance between the control and NMDA-treated groups.

9.     Fig. 2: Authors should present the results for CD3ζ-/- both in the presence and absence of NMDA to confirm the hypothesis of this study.

10. Fig. 3: Provide the data for CD3ζ-/- to confirm the mechanism.

11. Fig. 4: Explain why SU6656 and Saracatinib didn’t prevent RGC death when they can inhibit Src and Lck like PP2 and A419259. Explanations are warranted for the variable results.

12. Fig. 6: PP2 and A419259 inducing RGC death in CD3ζ-/- mice is confusing, as Figure 3, along with Figure 1, suggests there should be higher or similar RGC counts in CD3ζ-/- mice. This implies that SFK interdependent pathways are also strongly involved in RGC death.

13. Fig. 3F, 4E, and 6E: Authors used 10 samples for NMDA but 5 samples for other groups. The number of samples should be consistent to confirm the findings statistically significant.   

14. Fig. 7: Authors should present the results for CD3ζ-/- mice.

15. Table 1: Provide the rationale for using a higher concentration of SFK and Syk/Zap70 inhibitors than IC50.

16. Implications of NMDA toxicity to other retinal neurons not discussed.

Author Response

Response to R2

Comment  1. There is a remarkable diversity of RGCs, and the various subtypes have unique morphological features, distinct functions, and characteristic pathways that link the inner retina to the relevant brain areas. The central and peripheral regions of the retina have different numbers of ganglion cells. Provide details on how the consistency of RGC imaging between animals was maintained.

Response: It has been widely reported that the susceptibilities of RGCs to injuries vary significantly among different RGC types. The distribution of RGC types and density also vary with eccentricity and location of the retina (such as dorsal vs ventral and temporal vs nasal). Further, it is not clear whether the susceptibilities of RGCs to NMDA excitotoxicity vary at different retinal locations as that in glaucoma patients. Therefore, sampling the RGC density at any selective location of the retina under disease conditions might not precisely represent the responses of all RGC types of the whole retina.

To avoid the variations of RGC density sampling due to eccentricity and location-dependent changes from different mouse groups (WT and transgenic mice with or without injury), we sampled the RGC densities at 600 mm away from the center of the optic nerve head (304 mm x 304 mm, see Figure 1A for details) of four quarters (dorsal, ventral, temporal, and nasal) of all mice, quantified the RGC density of each quarter at this distance, and average the RGC density from all four quarters. As described in the “Material and Method” (section 2.5), this image acquisition and processing have been used in our previous studies [9,19,30,31]. Using this approach, our results represent the average RGC densities at 600 mm away from the center of the optic nerve head of four-quarters of all mice to minimize the potential effect of eccentricity and location-dependent changes of RGC density on our results.

Accordingly, we add the following paragraph in the Method (Section 2.5) in this revision to clarify how exactly the region was determined and selected for RGC quantification.

It has been widely reported that the susceptibilities of RGCs to injuries vary significantly among different RGC types. The distribution of RGC types and density also vary with eccentricity and location of the retina. To avoid the variations of RGC density due to eccentricity and retina location among different groups, we scan four stacks of images at four quarters of each retina at 600 mm away from the center of the optic nerve head of all mice (Fig 1A). Each stack covers 304 mm x 304 mm of the retina and the entire thickness of the ganglion cell layer (GCL) in wholemount retinas at intervals of 0.5 mm. The density of anti-RBPMS antibody labeled RGCs of each retina is averaged from the four stacks.

Comment  2. Provide the details of the number of animals, sections/animals, and region imaged within each section.

Response: As described in the response to comment 1, the regions of imaging are always the same place in all retinas, namely 600 mm away from the center of the optic nerve head (304 mm x 304 mm) of four quarters (dorsal, ventral, temporal, and nasal). The numbers of animals in each group are indicated in the legends of each figure or the text. The imaging depth always covers the entire GCL thickness.

Comment  3. RBPMS should label all the retinal ganglion cells? Thus, what is the cell type with only DAPI labeling?  Provide quantification of cells with only DAPI and discussion.

Response: The reviewer is right that RPBMS should label all RGCs. Therefore, the cells with only DAPI labeling in the GCL are displaced amacrine cells. Several previous studies have reported the percentage of RGCs and displaced amacrine cells in the mouse GCL. Using the anti-RBPMS antibody and NeuroTrace labeling, [135] reported that 42.8% of cells in GCL are RGCs. This is very close to the results reported by Pang and Wu (2011) who retrogradely labeled the RGCs by applying neurobiotin to the optic nerve stump and labeled 44% of cells in GCL as RGCs. Also, Jeon et al. (1998) quantified the number of axons in the optic nerve and total cells in the GCL and reported that 41% of cells in the GCL are RGCs and 59% of cells are displaced amacrine cells.

Comment  4. Line 94: The authors used 3.13 nmol NMDA, which does not match the concentration cited in the study (Reference 9), where the concentration is 6.25 nmol. Clarification is needed on how the concentration was determined for this study.

Response: In the study of reference 9, we used four different NMDA concentrations (0.325 mmol/L, 0.75 mmol/L, 3.125 mmol/L, and 6.25 mmol/L) to profile the susceptibility of four RGC types to NMDA excitotoxicity. Our data demonstrated that all of these four NMDA concentrations caused various amounts of RGC death. The concentration of 3.125 mmol/L, which is equivalent to 3.125 nmol, caused approximately 40-50% RGC death (also see reference 30), which leaves significant room for both further RGC death and rescue.

Accordingly, we add the following sentence in Section 3.1 in this revision to clarify how the concentration was determined for this study.

In our previous study, 3.13 nmol NMDA caused approximately 40-50% RGC death within 24 hours [30].

Comment  5. Authors should try different concentrations to determine the optimized NMDA concentrations.

Response: As described in the response to comment 4, we have already tried different concentrations to profile the susceptibility of RGCs to NMDA excitotoxicity in our previous studies and determine the concentration of 3.125 mmol/L ( 3.125 nmol) is an optimized NMDA concentration for the current study because it leaves significant room for both further RGC death and rescue (in case some inhibitors might cause additional RGC death with NMDA).

Comment  6. Fig. 1A: According to the text, the experiment was conducted for 24 hours, but the figure indicates an experiment of more than 8 days. Clarification is needed.

Response: In our study, the retinas were collected 24 hours after NMDA injection. The retinas are further processed after the tissue collection. Fig 1A illustrates the entire course of animal/tissue preparation including NMDA injection, retina collection, fixation, antibody incubation, tissue mounting, and imaging. The Fig 1A is further revised to clarify the process in this revision. Fig 1A is further revised to describe the entire process.

Comment  7. Fig. 1B: Authors should include separate panels for DAPI and anti-RBPMS, merging them in a separate panel for better understanding.

Response: The Fig 1 is revised accordingly.

Comment  8. Fig. 1C: RGC density is noticeably lower for CD3ζ-/- mice with NMDA, indicating that the CD3ζ mutation cannot protect RGCs from the effects of NMDA, which does not support the objective of this study. Although there appears to be a significant difference between WT and CD3ζ-/- mice, the values are very close (between 2000-3000). I suggest including more mice for a better outcome. Authors should also determine the statistical significance between the control and NMDA-treated groups.

Response: The Fig 1C is revised accordingly. In addition, we conducted a power analysis to determine whether the sample sizes were big enough for the comparison of the difference between WT and CD3ζ-/- mice with NMDA injection. The results show that the data with the sample sizes have 100% power to make the comparison. Per NIH guidelines for animal use in research, the minimum number of animals should be used for adequate analysis. Therefore, the number of mice used for this analysis should not be increased. The website address, the name of the analysis, and the results of the power analysis are attached below. The following paragraph is also added to the first paragraph of Section 3.1 of the revision based on this comment.

Added paragraph:

The RGC density of the WT retina (4947 ± 115 cells/mm2 (average ± SE)) is not different from that of CD3z-/- mice (5039 ± 140 cells/mm2, Student t-test, p = 0.679, Fig 1H) without NMDA injection. Figs 1F and 1G show representative images of the retinas of WT and CD3z-/- mice 24 hours after 3.13 nmol NMDA injection. The RGC density of the WT mice reduced from 4947 ± 115 cells/mm2 to 2540 ± 136 cells/mm2 (paired t-test, p = 0.0003, n = 5) and the RGC density of the CD3z-/- mice reduced from 5039 ± 140 cells/mm2 to 3380± 201 cells/mm2 (paired t-test, p = 7.17-5, n = 5) 24 hours after NMDA injection (Fig 1H). However, the RGC density of CD3z-/- mice with NMDA injection is significantly higher than that of WT mice with NMDA injection (2540 ± 136 cells/mm2 for WT vs 3380±201 cells/mm2 for CD3z-/- mice, Student t-test, p = 0.017, Fig 1H), which is 1.33 fold of WT mice. These results demonstrated that CD3z mutation partially but significantly increases RGC survival in NMDA excitotoxicity.

The website address for power analysis: ClinCalc.     https://clincalc.com/

Analysis name: Post-hoc Power Calculator.   https://clincalc.com/stats/Power.aspx

The power analysis results:

Comment  9. Fig. 2: Authors should present the results for CD3ζ-/- both in the presence and absence of NMDA to confirm the hypothesis of this study.

Response: Fig 2 shows the expression of SFKs in the retina of WT mice. This figure does not present any data related to NMDA application to CD3ζ-/- mice. However, we include additional data to show the expression of SFKs in CD3ζ-/- mice in Supplement Figure S2.

Comment  10. Fig. 3: Provide the data for CD3ζ-/- to confirm the mechanism.

Response: Fig 3 provided data for the search of SFKs which could reduce RGC death in NMDA excitotoxicity. The drugs used in this experiment (PP2 and A419259) are tested on CD3ζ-/- mice to confirm the mechanism in a later experiment presented in Fig 6.

Comment  11. Fig. 4: Explain why SU6656 and Saracatinib didn’t prevent RGC death when they can inhibit Src and Lck like PP2 and A419259. Explanations are warranted for the variable results.

Response: This question is specifically addressed by the analysis summarized in Fig 5. We conclude that SFK protects RGCs in NMDA excitotoxicity through inhibiting Hck, but not Src or Lck. Because neither SU6656 nor Saracatinib inhibits Hck and increases RGC survival, we conclude that Hck-mediated signaling is responsible for the increase in RGC survival.

Comment  12. Fig. 6: PP2 and A419259 inducing RGC death in CD3ζ-/- mice is confusing, as Figure 3, along with Figure 1, suggests there should be higher or similar RGC counts in CD3ζ-/- mice. This implies that SFK interdependent pathways are also strongly involved in RGC death.

Response: We agree with the reviewer that the results imply other signal pathways independent of SFK-mediated pathways are involved in RGC death. As briefly mentioned in the previous version of our manuscript (lines 332-334),  many SFK inhibitors mediate other pathways independent of SFKs [42–46]. PP2 and A419259 could not only protect RGCs in NMDA excitotoxicity by inhibiting the CD3ζ-Hck-Syk signal pathway shown in our study, but they can also inhibit pathways mediated by epidermal growth factor receptor (EGFR) or PKC to promote neuronal death (42–46). In WT mice, PP2 and A419259 inhibit both the CD3ζ-Hck-Syk signal pathway, which enhances RGC survival, and EGFR/PKC-mediated signal pathways, which reduce RGC survival. When the protective effect on RGCs by inhibiting the CD3ζ-Hck-Syk pathway overwhelms the effect mediated by EGFR/PKC pathways, these drugs generate a protective effect on RGCs. In CD3ζ-/- mice, the CD3ζ-Hck-Syk pathway is already inactivated before PP2 and A419259 application. The application of PP2 or A419259 will only inhibit the protective effect mediated by EGFR and PKC signaling and trigger additional RGC death. However, this possibility needs further study.

Comment  13. Fig. 3F, 4E, and 6E: Authors used 10 samples for NMDA but 5 samples for other groups. The number of samples should be consistent to confirm the findings as statistically significant.  

Response: In the experiments described in Figures 3, 4, and 6, we used one eye of the mice as NMDA control (NMDA only) and another eye as testing eye (NMDA + inhibitor). In data analysis, we lump the data of NMDA control eyes together as a control to compare with the test eyes. In response to this comment, we re-analyzed the data by comparing NMDA alone eyes with the testing eyes (NMDA+inhibitors) of each group without lumping them (NMDA alone) together and revised these figures accordingly.

Comment  14. Fig. 7: Authors should present the results for CD3ζ-/- mice.

Response: Revised accordingly.

Comment  15. Table 1: Provide the rationale for using a higher concentration of SFK and Syk/Zap70 inhibitors than IC50.

Response: We applied the concentration of the inhibitors to have the maximum effects in this study.

Comment  16. Implications of NMDA toxicity to other retinal neurons not discussed.

Response: We did not notice significant changes in the number of other retinal neurons after the NMDA injection. Because we collected the retinas 24 hours after NMDA injection, it is unlikely that the death of RGCs in our experiments would be the result of the death of other retinal neurons.

Reviewer 3 Report

Comments and Suggestions for Authors

The authors present the pharmacological dissection of the SKFs regulating Retinal Ganglion cell survival in response to NMDA-mediated toxicity in an experimental model of retinal degeneration in vivo.

The results are interesting and well presented. I have only two suggestions:

·      In page 6 line 200 the authors write “these results demonstrate that CD3 mutant prevents RGC death in NMDA exitotoxicity”.

However, clearly the rescue effect on CD3 KO is only partial. I realize that later in the article, this matter is addressed but I consider that, being this the first figure that the reader is analyzing, it would be important to use a precise language ( and for example, to indicate also if there is statistical significance between control and NMDA samples).

·      Similarly, it would be interesting to evaluate whether the differential expression of distinct SFKs or  down-stream signal transducers  could underlie RGC heterogeneity.

Are all the cell labeled equally? Is there a regionalization of the immunoreactivity? Maybe, this could enrich the Discussion section also.

Author Response

Responses to R3

Comment  1. On page 6 line 200 the authors write “these results demonstrate that CD3z mutant prevents RGC death in NMDA exitotoxicity”. However, clearly the rescue effect on CD3z KO is only partial. I realize that later in the article, this matter is addressed but I consider that, being this the first figure that the reader is analyzing, it would be important to use a precise language ( and for example, to indicate also if there is statistical significance between control and NMDA samples).

Response: We agree with this point. Accordingly, we significantly revised the whole paragraph for data presentation and ended the paragraph with the following sentence:

These results demonstrated that CD3z mutation partially but significantly increases RGC survival in NMDA excitotoxicity.

Comment  2. Similarly, it would be interesting to evaluate whether the differential expression of distinct SFKs or down-stream signal transducers could underlie RGC heterogeneity.

Response: This is a very interesting point. The most efficient approach to test this point would be genetic analysis of the expression of SFKs and their downstream molecules in various RGC subtypes. However, this is out of the scope of our current study.  Nonetheless, our immunostaining experiments indicate that all RGCs labeled by RPBMS are positive for SFKs (Fig 2, Src, Fyn, Lck, Yes, Hck) and Syks (Fig 7, Syk and Zap70) tested in this study.

Comment  3. Are all the cell labeled equally? Is there a regionalization of the immunoreactivity? Maybe, this could enrich the Discussion section also.

Response: To our best understanding, all RGCs are labeled crossing the entire retina by anti-RPBMS antibody and there is no regionalization of the immunoreactivity. This is supported by several previous studies that reported the percentage of RGCs and displaced amacrine cells in the mouse GCL. Using the anti-RBPMS antibody and NeuroTrace labeling, [135] reported that 42.8% of cells in GCL are RGCs. This is very close to the results reported by Pang and Wu (2011) who retrogradely labeled the RGCs by applying neurobiotin to the optic nerve stump and labeled 44% of cells in GCL as RGCs. Also, Jeon et al. (1998) quantified the number of axons in the optic nerve and total cells in the GCL and reported that 41% of cells in the GCL are RGCs and 59% of cells are displaced amacrine cells. We also show in Fig 1B (new) that retinal neurons are equally labeled by anti-RPBMS antibodies crossing the entire retina.

In addition, we confirmed the NMDA distribution inside the eye after the intraocular injection by co-injecting NMDA with Alexa FluorTM 488 conjugated Cholera Toxin Subunit B (CTB, 0.2%, Cat #: C22843, Thermo Fisher Scientific, Eugene, OR, United States). We included a representative image of a flat-mount retina labeled by Alexa FluorTM 488 conjugated CTB in this revision (Fig 1C (new)) demonstrating the homogenous distribution of Alexa FluorTM 488 conjugated CTB labeled cells in the injected eyes.
